# Electrochemical Measurement of Interfacial Distribution and Diffusion Coefficients of Electroactive Species for Ion-Exchange Membranes: Application to Br_2_/Br^−^ Redox Couple

**DOI:** 10.3390/membranes12111041

**Published:** 2022-10-26

**Authors:** Dmitry V. Konev, Olga I. Istakova, Mikhail A. Vorotyntsev

**Affiliations:** 1Federal Research Center for Problems of Chemical Physics and Medicinal Chemistry of the Russian Academy of Sciences, Chernogolovka 142432, Russia; 2Frumkin Institute of Physical Chemistry and Electrochemistry of the Russian Academy of Sciences, Moscow 119071, Russia

**Keywords:** membrane electrode assembly, crossover of molecule/co-ion, co-ion distribution coefficient between membrane and solution, solute electroactive component, voltammetric and chronoamperometric techniques, diffusion permeability of membranes, diffusion coefficient inside membrane, bromide anion, bromine

## Abstract

A novel method has been proposed for rapid determination of principal transmembrane transport parameters for solute electroactive co-ions/molecules, in relation to the crossover problem in power sources. It is based on direct measurements of current for the electrode, separated from solution by an ion-exchange membrane, under voltammetric and chronoamperometric regimes. An electroactive reagent is initially distributed within the membrane/solution space under equilibrium. Then, potential change induces its transformation into the product at the electrode under the diffusion-limited regime. For the chronoamperometric experiment, the electrode potential steps backward after the current stabilization, thus inducing an opposite redox transformation. Novel analytical solutions for nonstationary concentrations and current have been derived for such two-stage regime. The comparison of theoretical predictions with experimental data for the Br_2_/Br^−^ redox couple (where only Br^−^ is initially present) has provided the diffusion coefficients of the Br^−^ and Br_2_ species inside the membrane, D(Br^−^) = (2.98 ± 0.27) 10^−6^ cm^2^/s and D(Br_2_) = (1.10 ± 0.07) 10^−6^ cm^2^/s, and the distribution coefficient of the Br^−^ species at the membrane/solution boundary, K(Br^−^) = 0.190 ± 0.005, for various HBr additions (0.125–0.75 M) to aqueous 2 M H_2_SO_4_ solution. This possibility to determine transport characteristics of two electroactive species, the initial solute component and its redox product, within a single experiment, represents a unique feature of this study.

## 1. Introduction

In rechargeable chemical power sources such as redox flow batteries (PRBs), which use electroactive components dissolved in the electrolytes of the positive and/or negative electrodes, the crossover, i.e., the problem of their transmembrane penetration into the opposite chamber, is of particular importance [1,2]. In addition to a decrease in the battery’s capacitance as a result of redox reactions of the penetrating reagent with “local” substances, the crossover can cause poisoning of the electrode surface and the associated slowdown in the target half-reaction, as well as lead to a change in the composition of electrolytes which seriously affects the characteristics and service life of the device as a whole [3,4]. Therefore, the crossover of redox components through the membrane, for example, in relation to vanadium PRBs, should be considered in the course of simulations of the functioning of both individual membrane-electrode assemblies [5,6,7] and their stacks [8,9]. Both novel membrane materials [10,11,12] and novel methods for modification of membranes [13,14,15] are studied to minimize these harmful effects.

This problem results in the necessity to experimentally determine the parameters characterizing the rate of this phenomenon for various membranes and electroactive components under model conditions. For its solution, so-called H-cells are most widely used [16,17,18,19], where the transported component is initially present only in one of two electrolytes in the chambers which are separated by the membrane sample of a known area inside a relatively narrow channel. The crossover leads to an increase of the concentration of the component in the receiving chamber which initially did not contain it and its variation is monitored with the use of a suitable analytical procedure, e.g., electrochemical methods (potentiometry [16], amperometry [16,17]) or other techniques such as densitometry [18] and UV-Vis spectroscopy [19].

The resulting dependence of its concentration in time is simulated based on a model for the component’s transport across the membrane. This treatment provides specific characteristics of the crossover, such as the value of the mass transfer coefficient, or of the diffusion coefficient of the component inside the membrane, or of the diffusion flux/current density. Among the disadvantages of this method, one should indicate the long duration of measurements, the complexity of data processing due to the need to consider the time-varying difference in the concentrations of the component on both sides of the membrane, as well as the problem of considering the distribution of the component at each of two membrane/solution interfaces.

This method was modified in [17,20,21,22,23], where the electrochemical method was used to directly measure the flux of the substance through the membrane, instead of tracing its concentration accumulated in the receiving chamber. For this purpose, a porous (e.g., carbon) [17,20] or metal mesh [16,21,22] working electrode was placed inside the receiving chamber of a two-chamber cell in contact with the membrane surface so that the diffusion flow of the substance under study through the membrane is converted into the electric current. This electrode is polarized with the use of the auxiliary and reference electrodes located on the same side of the membrane. Thus, the study of the crossover is carried out without imposing the electric field across the membrane. In particular, in [16,17], such measured data for the diffusion flux for mixed bromine–bromide solutions under the steady-state and nonstationary regimes were recalculated in terms of the values of the diffusion coefficient and the membrane permeability for this system

A characteristic feature of such measurements of the diffusion permeability of membranes [17,20,21,22,23] is the preliminary equilibration of the membrane with the *background* solution (e.g., aqueous NaBr) inserted into both chambers of the cell. Then, this solution in the source chamber is replaced as quickly as possible with the solution containing the electroactive component and the same background electrolyte (e.g., NaBr + Br_2_). At the same time, the working electrode located in the receiving chamber (inside the solution or at the membrane surface) is polarized up to the potential value corresponding to the electrochemical transformation of the electroactive component on the electrode surface under the diffusion-limited regime after its passage through the membrane; then, the time-variation of the current is recorded, which reflects the increase of the flux of the substance through the membrane after the solution change in the source chamber.

Interpretation of such data for the nonstationary current is based on the model that assumes that the equilibrium concentration of the electroactive substance in the surface layer of the membrane near its boundary with the source solution is established instantly, as soon as the substance under study appears in this solution (“moment of the beginning of the process”). During the subsequent time interval, it is spreading inside the membrane via the nonstationary diffusion mechanism. The data fitting is performed with the use of an approximate analytical formulae for the flux of the substance across the membrane as a function of time, which assumes that the transport starts at a certain moment, t = 0, when the substance appears inside the surface layer of the membrane. The procedure identifies this moment, t = 0, with the “moment of the beginning of the process”, even though the former is obviously *later* than the latter. In addition, the transport model assumes that the concentration of the substance inside the surface layer of the membrane near its second boundary remains always equal to zero due to its discharge at the porous or mesh electrode located outside, while the electrochemical reaction at such an electrode occurs inside the *layer of a finite thickness* because of a nonzero diffusion transport resistance outside the membrane.

Another variant of experimental studies of transport rates of solution components through a permeable film uses the stationary or rotating disk electrode coated by a thin layer of the material (e.g., Nafion) deposited on its surface from solution of its precursor [24,25,26,27,28]. Such a modified electrode is immersed into an electrolyte solution containing the electroactive component under study. After applying the appropriate potential, the steady-state or relaxational current is measured. As a rule, such systems are designed to study the effect of the deposited material on the kinetics of electrode reactions of this electroactive component. Comparison of the transport characteristics of deposited layers with the permeability of prefabricated membranes (even those composed of the same material) is usually not carried out in view of the expected difference in the parameters of the transporting pores of the membrane and of the deposited layer because of the non-identical methods used for their syntheses. In addition, the thicknesses of the deposited layers are mostly significantly less than those for membranes made via a special technological process, and the uniformity of their thicknesses is much less controlled. As a result, the results obtained by the method where the film is deposited at the electrode from its precursor are hardly applicable for estimating the diffusion permeability of the membrane of the same composition.

In this work, we have proposed a novel approach for measuring the diffusion coefficient of a solute redox active substance inside the membrane and its distribution coefficient at the membrane/solution interface, as well as the diffusion coefficient of the electrochemical product of this substance inside the membrane. The method is based on direct measurement of the current density due to the discharge of the substance on the surface of a compact (non-porous) electrode, which is *mechanically coated by the membrane*, separating it from the solution containing the substance.

Analogously modified electrodes (with a film-coated surface) are used in analytical practice, mainly as an element of the Clark sensor [29,30,31,32], in which a film pressed to the electrode surface standardizes the mass transfer of the analyzed component, thus ensuring the reproducibility of its amperometric determination. To our best knowledge, such an approach has never been used previously for the analysis of the permeability of ion-exchange membranes with respect to electroactive components from an external solution.

The feasibility of the proposed method has been evaluated in relation to the transport of bromide anions and bromine molecules across a perfluorinated sulfonate cation-exchange membrane from aqueous sulfuric acid solutions containing various amounts of hydrobromic acid. The choice of this transporting electroactive component, Br^−^, for testing the proposed method is due to the broad use of the bromine/bromide redox couple in power sources, in particular in PRBs of various types [6,33,34,35,36,37,38,39,40,41]. We carried out studies of hydrogen-bromate flow batteries where the entering electrolyte of the positive electrode functions represents an aqueous solution of a bromate–sulfuric acid mixture [39,40]. Bromate reduction proceeds inside the cell via *redox-mediator autocatalysis* by the Br_2_/Br^−^ redox couple, which results in an increase of the concentrations of its components from trace amounts at the entrance to the cell up to extremely high ones (comparable to the initial bromate concentration). Thus, HBr + H_2_SO_4_ mixed solutions of various concentrations chosen for our primary crossover studies represent a simplified example of the electrolyte inside such a power source.

The presence of a fixed and relatively high concentration of sulfuric acid (compared to the variable amount of HBr) provides the important extra advantages that protons are the only counterion species inside the membrane and their concentration therein is practically constant for the series of HBr concentrations. These factors ensure the suppression of migration effects for the bromide transport inside the membrane as well as a good proportionality between the equilibrium Br^−^ concentrations in the membrane and solution, etc.

## 2. Materials and Methods

Electrochemical measurements were carried out under inert atmosphere of Ar (99.999%, Linde Gas Rus, Russia) in a conventional three-electrode cell (solution volume: 20 mL) without separation of electrolyte space on an Autolab 302N potentiostat (Metrohm, the Netherlands). Large-area platinum foil was used as a counter electrode to pass a high-density current through the working electrode. The reference electrode was a two-chamber (double frit) Ag/AgCl/KCl (saturated) electrode (its potential: 0.198 V vs. SHE), and solution in the intermediate chamber was aqueous 2 M H_2_SO_4_ (the same acid concentration as that in the working electrode solution understudy). All potential values below are provided vs. this reference electrode.

A special home-made design was used for the working electrode (Figure 1), where the electrode surface represented a circular end face of platinum rod 5, which was fixed inside insulating body 2 together with current supplier 1. The electrode surface was separated by membrane 7 (held in tight contact with the electrode surface by cover 4) from the external solution.

For their tighter contact, the shapes of the surfaces of both the electrode and of the insulating body were made slightly spherical by their grinding and subsequent polishing with abrasive material attached to watch glass. Before fixing the membrane, the rounded surface of the end face of body 2 (to be pressed against the membrane surface together with platinum rod 5) was subjected to final polishing with diamond suspension (particle size: 0.25 µm, ECIL, France).

Membrane thickness was measured after equilibration in contact with each electrolyte solution. Three Nafion 212 membrane samples were kept in 2 M H_2_SO_4_ solutions containing 0.125, 0.5, or 1 M HBr; then, the thickness was determined with a micrometer. The measurement results were 57, 58, and 59 µm (results of three measurements of the thickness of each sample were averaged), respectively. The observed small increase in the membrane thickness is within the scatter of the measurement result for an individual sample. Therefore, its average value, 58 μm, was used for subsequent calculations for the whole range of HBr concentrations. 

In the course of the electrode preparation for the experiments, each sample of the Nafion 212 membrane (DuPont, USA), preliminarily cut in the form of a circle (diameter: 7.5 mm), was subjected to the procedure described in Table 1 of [42] for the “boiled pretreatment process”; then, it was kept for at least half an hour in contact with the electrolyte solution of the corresponding composition: 2 M H_2_SO_4_ + certain HBr concentration.

After this treatment, drops of solution were removed from its surface. Then, it was rigidly fixed inside the assembly of elements 3, 4, and 6 (Figure 1), after which the assembly was screwed onto element 2 (with electrode 5 pressed into it) to achieve a tight contact of the platinum electrode surface with the membrane. The quality of their contact was controlled visually by the displacement of air from the membrane/electrode contact zone, which was accompanied to a noticeable increase in the transparency of the interface when observed in reflected light. At the same time, excessive tension of the membrane was prevented by insertion of a spacer-limiter (not shown in Figure 1) of a pre-calculated thickness into the gap between elements 2 and 3. The whole membrane-fixation procedure was performed within a time duration shorter than a minute to exclude changes in the membrane thickness due to water evaporation. Immediately after this stage, the prepared working electrode–membrane assembly was immersed into the electrolyte solution and was in contact with it during all stages of subsequent electrochemical measurements.

For determination of the main parameters of the transmembrane transport via electrochemical measurements, we used the method of potentiostatic chronoamperometry for the diffusion-limited current regime (see below, the section on theoretical modeling of this procedure). To choose the appropriate range of potentials for this technique, stationary and cyclic voltammograms were measured for the membrane-coated platinum working electrode of the construction described above (Figure 1). Measurements of stationary voltammograms were carried out by stepwise changes of potential, keeping each potential value constant until a steady-state current was reached (10 to 30 s). During measurements, the electrolyte solution was agitated by a magnetic stirrer. Cyclic voltammetry was performed under the same conditions as the stationary one, and the potential sweep rate was varied in the range from 0.03 to 3 V/s.

Potentiostatic chronoamperometry of the membrane-coated electrode in contact with each H_2_SO_4_ + HBr solution was carried out according to a special procedure, which included the following steps: (a)The electrode was maintained under the potentiostatic regime at E = 0.4 V until the stationary current was reached (to remove traces of molecular bromine from the membrane and the near-electrode solution).(b)The electrode was maintained under the open-circuit potential (OCP) regime for several tens of seconds, or until a stationary OCP value was reached.(c)The potential was changed stepwise to the constant value corresponding to the bromide-ion oxidation (2 Br^−^ − 2 e^−^ = Br_2_) on platinum under the diffusion-limited current regime (this value was determined in the preliminary stationary voltammetry experiment described above). Registration of the current decay in time after the potential step (these data are called the “current transient” below) was performed for several different time ranges after the potential step: each 10^−4^ s during the first second, then each 10^−3^ s during the range from 1 to 10 s after the step and each 10^−2^ s after 10 s until the steady-state current was established. The total duration of stage (c) was from 60 to 120 s.(d)After the current stabilization over the course of stage (c), the second potential step was imposed to the value corresponding to the bromine-molecule reduction (Br_2_ + 2 e^−^ = 2 Br^−^) on platinum under the diffusion-limited current regime. Registration of the passing current was carried out in the same way as that of stage (c).

The set of stages (a)–(d) was repeated three times, and the duration of stage (b) over the course of these repetitions increased in the series: 60, 240, and 360 s. The current transients registered during steps (c) and (d) were processed with the use of the theoretical model described below.

For the sake of interpretation of the characteristic segments of current transients, similar measurements were taken with the use of a *non-coated* platinum electrode of the same area (circular section of a wire of 1 mm in diameter inside a glass insulator), i.e., with its surface being directly in contact with the same electrolyte.

## 3. Theory of Transport Processes Inside Membrane

### 3.1. Description of the System

A working electrode coated by a *cation-exchange* membrane (described in the previous section, Figure 1) was immersed into an electrolyte solution which also contains electroactive component B of concentration c^0^, which is able to participate in a chemically reversible electron-transfer transformation at the electrode surface (this term means that the electrochemical transformation of B species into its product, C, may be reversed back into the same B species by means of imposing a sufficiently high backward overpotential). The membrane-coated working electrode was constructed in such a way so that it allows both the reacting species, B, and the reaction product, C, to pass through the membrane in the opposite directions.

An experimental study has been performed for the aqueous 2 M H_2_SO_4_ solution with the addition of various amounts, c^0^, of the redox-active component, HBr (concentration range: from 0.125 to 1 M). The bromide anion, B, may cross the solution/membrane boundary, then pass across the membrane to be oxidized into bromine, C, at the electrode surface, while the generated bromine may be reduced back to bromide anion, depending on the electrode potential:2 X^−^ − 2 e^−^ ⇄ X_2_(1)

One should pay attention that this process does not correspond to the conventional type of *unity stoichiometry reactions*: Red − n e^−^ = Ox, which is mostly considered. To include both types of reactions, the theoretical relations below were derived for the most general stoichiometry of the redox process at the electrode:n_B_ B − n e^−^ = n_C_ C(2)
while for the particular case of reaction (1), halide anion, X^−^, and halogen molecules, X_2_, are denoted as B and C, respectively, n_B_ = 2, n_C_ = 1, and n = 2.

There are no significant concentration gradients in solution; in particular, the concentrations of the halide ion, X^−^, and of the halogen, X_2_, in solution near the interface with the membrane are equal to their bulk solution concentrations, c^0^ and 0, respectively, owing to the solution agitation or/and slow transmembrane transport of X^−^ and X_2_ species.

The solution and the electrode are separated by the membrane (Figure 1), which is modeled as a uniform plane layer of a thickness, L. All solute components as well as the reaction product, X_2_, can penetrate into the membrane so that their concentrations in the course of nonstationary experiments depend on the spatial coordinate, x, normal to the electrode surface, and on time, t.

It is assumed that the X^−^ concentration inside the membrane, B(x,t) for 0 < x < L, is much lower than the proton one (which is determined primarily by the fixed-charge density inside this medium) due to both the *coion*-type of X^−^ and its relatively small concentration in solution, c^0^.

It is also supposed that owing to a rapid interfacial exchange across the membrane/solution boundary, the electrochemical potentials of each component at both sides of the boundary are equal to each other. Then, the concentrations of the halide ion, X^−^, and of the halogen molecules, X_2_, inside the membrane near its interface with the solution, B(L,t) = B* and C(L,t), respectively, are proportional to their bulk solution concentrations, c^0^ and 0:B(L,t) = B* = K c^0^, C(L,t) = 0(3)
for any moment of the whole procedure, where the value of K (dependent on the solution composition, first on its pH) is small compared to 1, due to the co-ion type of species B (X^−^) and a high bulk solution proton concentration.

During the pretreatment period, t < 0, the electrode is in the OCP regime, i.e., the current, I, is equal to 0. If its duration is sufficiently extended, all concentrations are uniformly distributed across the membrane, in particular the X^−^ and X_2_ concentrations are equal to B* and 0, respectively.

After the pretreatment period, the electrode potential is shifted in the positive direction, thus inducing the oxidation of species B (X^−^) at the electrode surface into species C (X_2_).

### 3.2. Cyclic Voltammetry and Potentiostatic Chronoamperometric Procedures

Two nonstationary electrochemical regimes have been employed in this study.

First, the electrode potential changes in a cyclic manner as a linear function of time between two limits, E_i_ and E_f_, where the reaction rate is practically zero at its former value, E_i_, while the electron-transfer process proceeds under transport-limited control for its latter value, E_f_. For the Br^−^/Br_2_ couple, the potential sweep limits were chosen as E_i_ = −0.15 V and E_f_ = 1.2 V based on stationary voltammetry data (see below). The potential scan rate was varied between 0.03 and 3 V/s.

Second, amperometry under potentiostatic conditions has been used. After a pretreatment period (where the potential corresponds to a very low current), the electrode potential changes in the stepwise manner to a sufficiently positive value, E_fB_, to induce such a rapid transformation of the solute electroactive species, B, at the electrode surface so that its local concentration near the electrode surface, B(0,t), drops to a very low value. This potential, E_fB_, is kept for the whole relaxation period when the nonstationary concentration distributions of species B and C inside the membrane change towards their steady-state profiles. After reaching this stationary state of the system, the electrode potential changes abruptly in the opposite direction up to its value, E_fC_, thus inducing a backward transformation of the reaction product, C, into the initial species, B, also under the potentiostatic transport-limited regime.

For the Br^−^/Br_2_ redox couple, the potentials of both stages have been chosen in conformity with the value of its formal potential of the X^−^/X_2_ redox couple, 0.89 V vs. our reference electrode. Correspondingly, during the initial pretreatment regime, electrode potential was kept at 0.4 V for 100 s; after this period, the OCP (zero current regime) was imposed for a period of 60, 240, or 360 s; then, the potentiostatic Br^−^ oxidation stage was switched on (E_fB_ = 1.0 V for 120 s), and finally, the Br_2_ reduction stage took place, also under the potentiostatic regime (E_fC_ = 0.4 V for 120 s).

### 3.3. Equations for Treating Experimental Data

If the X^−^ concentration inside the membrane, B(x,t) for 0 < x < L, is much lower than the proton one, the latter plays the role of a background electrolyte, and one may disregard the migrational contribution to the X^−^ transport across the membrane, compared to the diffusional one. Then, the concentration distributions of X^−^ and X_2_ species may be described by the nonstationary Fick equation:∂B/∂t = D_B_ ∂^2^B/∂x^2^, ∂C/∂t = D_C_ ∂^2^C/∂x^2^  for 0 < x < L(4)
where D_B_ and D_C_ are the diffusion coefficients of species B (X^−^) and C (X_2_), respectively, inside the membrane region.

Initial and boundary conditions depend on the regime of the electrode potential variation.

### 3.4. Cyclic Voltammetry for the X^−^/X_2_ Redox Couple

Electrode potential is subjected to its linear variation in time: E(t) = E_i_ + v t, starting from its initial value, E_i_, at t = 0, where the potential is much more negative than the formal potential of the X^−^/X_2_ redox couple (0.85 V vs. our reference electrode) so that the X^−^ concentration at the electrode/membrane interface, B(0,t), remains close to the initial value, B* = K c^0^, and the passing current is very low. When the potential, E(t), becomes sufficiently positive, it induces a more and more rapid oxidation of X^−^ into X_2_ at the electrode surface (Equation (1)), thus inducing a progressive diminution of its surface concentration, B(0,t), accompanied by the concentration wave propagation inside the membrane. At the largest potential of the forward scan, E_v_ = E_i_ + v t_v_, where the direction of the potential sweep is inversed, the surface concentration, B(0,t), is already very small compared to B* = K c^0^. Over the course of the backward sweep, the potential varies in the opposite direction: E(t) = E_v_ − v (t − t_v_) = E_i_ − v (t − 2 t_v_), so that the product of the forward reaction, X_2_, is reduced back into X^−^.

Both the forward and backward branches of the voltammogram represent curves possessing a maximum of the current. Within the raising region of the forward scan, the interfacial concentration inside the membrane, B(0, t), is close to its initial value, B* = K c^0^, while the surface concentration of the product, X_2_, increases exponentially: C(0,t) ~ exp (nFE/RT), where n = 2, and the diffusion layer thickness is constant: δ ~ (D_C_ RT/nFv)^1/2^, i.e., very small compared to the membrane thickness, L, if the scan rate, v, is not especially small (see below). Progressive extension of the diffusion layer takes place after the passage of the anodic current peak, I_pa_: δ ~ [D_B_ (t − t_pa_)]^1/2^ = [D_B_ (E − E_pa_)/v]^1/2^. Diffusion coefficients of the reactant and of the product, D_B_ and D_C_, are of the order of 10^−6^ cm^2^/s (see results below), and the potential interval is about a few hundred mV, while the thickness, L, of the membrane under study is around 58 µm. Then, δ is very small compared to the membrane thickness, L, if v is much larger than *a few tens of mV/s*. Thus, one may expect that the whole perturbed concentration region (i.e., the nonstationary diffusion layer) is deeply inside the membrane space so that one may apply the cyclic voltammetry theory for the nonstationary *semi-infinite* diffusion.

The halide-to-halogen transformation (Equation (1)) at the Pt electrode represents a relatively rapid process [43,44]. It implies that one may apply results of the recent study of reversible electrode reactions of a non-unity stoichiometry by Gómez-Gil, Laborda, and Molina [45]. System (1) corresponds to the 2:1 type for n = 2 so that the forward peak potential, E_pa_, is to move in the negative direction as a function of the initial X^−^ concentration inside the membrane, and the peak-to-peak separation should be about 43 mV for any values of the scan rate, v, and of the initial concentration of the species inside the membrane, K c^0^, while the peak current for the forward scan is given by Equation (5):I_pa_ = Ψ_p_ FA K c^0^ (D_B_ Fv/RT)^1/2^, where Ψ_p_ = 0.177 n^3/2^ = 0.501(5)

Here, D_B_ is equal to the diffusion coefficient of X^−^ species *inside the membrane* (Equation (4)).

### 3.5. Chronoamperometry: Stage of B-to-C Reaction (X^−^ Oxidation for Scheme (1))

Similar to the cyclic voltammetry study (see above), the system is subjected to an initial equilibration, i.e., to a long-time exposure under the OCP regime (I = 0 for t < 0), so that the concentration of the B (X^−^) species is constant, while there are no C (X_2_) species: B(x,t) = B* = K c^0^, C(x,t) = 0 for t ≤ 0, 0 < x < L (6)

Then, the potential is subjected to a large-amplitude step toward a sufficiently positive value (well over the formal potential of the B/C (X^−^/X_2_) redox couple, 0.89 V vs. our reference electrode) so that the B (X^−^) concentration at the electrode/membrane interface, B(0,t), becomes close to zero, and this condition remains valid for the whole time interval of this stage, i.e., until the current stabilization:B(0,t) = 0 for t > 0 (7)

Reactional scheme (2) provides relations between the passing current, I, and the interfacial fluxes of species B and C inside the membrane at x = 0, J_B_ and J_C_, respectively:I/nFA = −J_B_/n_B_ = J_C_/n_C_, where J_B_ = −D_B_ ∂B/∂x, J_C_ = −D_C_ ∂C/∂x at x = 0, t > 0(8)

F and A are the Faraday constant and the membrane surface area, and n, n_B_, and n_C_ are the stoichiometric coefficients of electrons, species B and C, respectively, in Equation (2), where n = 2, n_B_ = 2, and n_C_ = 1 for the X_2_/X^−^ redox couple.

Nonstationary concentration distribution of B (X^−^) species, B(x,t), satisfies Equation (4) with initial condition (6) and boundary conditions (3) and (7). Its Laplace transform, B_p_(x), represents a solution of Equation (9):d^2^B_p_(x)/dx^2^ = p B_p_(x) − K c^0^, B_p_(0) = 0, B_p_(L) = K c^0^/p (9)
where,
(10)$$B_{p}(x)=\int_{0}^{\infty }B(x,t)\;\exp\;(-\mathrm{pt})\;\mathrm{dt}$$

At the end of the relaxation period, the concentration distribution and the current approach their steady-state values:B(x,t) → B_ss_(x) = B* x/L = K c^0^ x/L,I(t) → I_ss_ = (n/n_B_) FA D_B_ K c^0^/L for t → ∞(11)
where n and n_B_ are the stoichiometric coefficients of reaction (2), and n/n_B_ = 1 for reaction (1).

The solution of Equation (9) gives an *exact* expression for the nonstationary concentration distribution, Equation (12), where the steady-state contribution, B_ss_(x), is defined by Equation (11):(12)$$B(x,t)=B_{\mathrm{ss}}(x)+2\;B^{*}\;\sum_{n=1}^{\infty }(\pi n){}^{-1}\;\sin\;(\pi n\;x/L)\;\exp\;(-n^{2}\;\pi^{2}\;D_{B}\;t/L^{2})$$

It results in the *exact* expression (10) for the nonstationary current during this stage:(13)$$I_{B}(t)=I_{\mathrm{ss}}\;[1+2\;\sum_{n=1}^{\infty }\exp\;(-n^{2}\;\pi^{2}\;D_{B}\;t/L^{2})]\;$$
where the steady-state current, I_ss_, is defined by Equation (11). Equation (10) demonstrates that the current, I_B_(t), approaches its steady-state value, Equation (11), for sufficiently extended time moments. Their difference, I_B(ns)_(t), given by Equation (14), diminishes exponentially, as in Equation (15):(14)$$I_{B(\mathrm{ns})}(t)\equiv I_{B}(t)-I_{\mathrm{ss}}=2\;I_{\mathrm{ss}}\;\sum_{n=1}^{\infty }\exp\;(-n^{2}\;\pi^{2}\;D_{B}\;t/L^{2})$$
 I_B(ns)_(t) ≅ 2 I_ss_ exp (−π^2^ D_B_ t/L^2^)(15)
for the sufficiently large time range.

Integration of the *exact* expression (14) for this current, I_B(ns)_(t), within the whole duration of this stage, gives the “nonstationary charge”, Q_B(ns)_:(16)$$Q_{B(\mathrm{ns})}=\int_{0}^{\infty }I_{B(\mathrm{ns})}(t)\;\mathrm{dt}=(1/3)\;I_{\mathrm{ss}}\;L^{2}/D_{B}=(1/3)\;(n/n_{B})\;\mathrm{FA}\;K\;c^{0}\;L$$

Relation (16) contains expressions for the principal parameters of stage 1, D_B_ and K, via experimentally measurable quantities:K = Q_B(ns)_ [(1/3) (n/n_B_) FA c^0^ L]^−1^;  D_B_ = L^2^ I_ss_/3 Q_B(ns)_(17)

Application of the Poisson summation formula allows one to derive another *exact* solution for the current, I_B_(t), from Equation (10):(18)$$I_{B}(t)=I_{\mathrm{ss}}\;L\;(\pi t\;D_{B}){}^{-1/2}\;[1+2\;\sum_{k=1}^{\infty }\exp\;(-k^{2}\;L^{2}/D_{B}\;t)]$$

Within the relatively short time range, it gives the Cottrell formula for the current:I_B_(t) ≅ I_B(CC)_(t) = CC_B_ t^−1/2^(19)
for sufficiently small time values, where: CC_B_ = I_ss_ L (π D_B_)^−1/2^ = (n/n_B_) FA K c^0^ (D_B_/π)^1/2^
(20)

Approximate expression (19) for I_B_(t) is valid with an *exponential precision* for small time values, i.e., the difference between the exact and approximate formulae, Equations (18) and (19), is of the order of 2 I_B(CC)_(t) exp (−L^2^/D_B_ t).

Thus, the use of experimental data for this X^−^ oxidation stage allows one to extract the values of I_ss_ and Q_B(ns)_ as well as the dependences: I_B(CC)_(t) and I_B(ns)_(t), for the short or extended time ranges, respectively. Then, one can determine two independent parameters, e.g., FA K c^0^ L and L^2^/D_B_ or FA K c^0^ D_B_^1/2^.

Equations (5) and (20) show that the voltammetric peak current, I_pa_, and the Cottrellian constant for species B (X^−^), CC_B_, contain the same combination of unknown parameters, K and D_B_, so that their ratio for the X^−^ oxidation via Equation (1) is given by Equation (21):I_pa_/CC_B_ = Ψ_p_ π^1/2^ (Fv/RT)^1/2^ where Ψ_p_ π^1/2^ = 0.887(21)

For other stoichiometric coefficients in reaction (2), the proportionality between I_pa_ and CC_B_ for various scan rates, v, is retained, but with different values of the proportionality coefficient.

Equation (21) implies that this ratio of experimentally measurable quantities, I_pa_/CC_B_, should only depend on known parameters, being independent of the membrane thickness, L, halide distribution coefficient, K, or of the diffusion coefficient of X^−^ inside the membrane, D_B_. Even though the experimental data for a set of systems below give values of the ratio, I_pa_/CC_B_, which are in a good agreement with predictions of Equation (21), there are inevitable experimental effects (incomplete homogeneity of membrane, finite thickness of membrane compared to the nonstationary diffusion layer thickness, insufficiently large value of the ratio of counterion-to-co-ion concentrations inside the membrane, etc.), which lead to deviations from Equation (21).

It is the reason why it is useful to recalculate experimentally found values of the peak current, I_pa_, for each concentration of HBr in the external electrolyte solution and each scan rate in terms of an equivalent experimental parameter, CCV, via Equation (22): CCV ≡ I_pa_ [Ψ_p_ π^1/2^ (Fv/RT)^1/2^]^−1^ = 1.127 I_pa_ (RT/Fv)^1/2^(22)

If Equations (5) and (20) *were* precisely fulfilled, then the values of the CCV parameter for each HBr concentration *were* independent of the scan rate and equal to the value of the CC_B_ parameter for the same concentration. In other words, their non-equality characterizes the degree of inaccuracy of these theoretical equations to describe the process under study. Our treatment of experimental data below provides an illustration of these aspects.

According to Equation (20), CC_B_ is determined by the product of other parameters of the system: A K c^0^ D^1/2^, while Equation (11) for I_ss_ contains their different combination: A D K c^0^/L. Assuming that estimates for the values of the electrode surface area, A, the bulk-solution concentration of X^−^, c^0^, and the membrane thickness, L, are available, one can express for reaction (1) the diffusion coefficient of X^−^ species inside the membrane, D_B_, and its distribution coefficient at the membrane/solution interface, K, via measurable quantities, CC_B_ and I_ss_:D_B_ ≅ π^−1^ (L I_ss_/CC_B_)^2^, K ≅ π (FA c^0^ L)^−1^ CC_B_^2^/I_ss_(23)

### 3.6. Chronoamperometry: Stage of Backward C-to-B Reaction (X_2_ Reduction for Scheme (1))

The previous B-to-C transformation (X^−^ oxidation) stage terminates when the steady state has been achieved, i.e., the concentration distribution of B (X^−^) species and the current have approached their expressions (11), B_ss_(x) and I_ss_. In parallel, the distribution of the reaction product, C (X_2_), also reaches a time-independent limit:C_ss_(x) = C* (1 − x/L), (24)
C* = C_ss_(0) = (n_C_/n_B_) B* D_B_/D_C_ = (n_C_/n_B_) K c^0^ D_B_/D_C_, where n_C_/n_B_ = ½ for reaction (1).

Here, C_ss_(L) = 0, because of the absence of X_2_ species in the external solution and of the equilibrium exchange across the membrane/solution interface, and n_C_/n_B_ = ½ for reaction (1). The starting concentration at the membrane/electrode boundary, C*, is directly related to the steady-state current prior to the second potential step, Equation (11), where n/n_C_ = 2 for reaction (1):I_ss_ = (n/n_C_) FA D_C_ C*/L(25)

Then, electrode potential makes the second stepwise change, this time to a sufficiently negative (more accurately: less positive) value to induce a rapid transformation of C (X_2_) species into B (X^−^) ones (Equation (2)) so that the C (X_2_) concentration at x = 0 drops abruptly to its zero value:C(0,t) = 0  for t > 0(26)

For the sake of simplification of the expressions, below, this moment of the second potential step is denoted again as t = 0.

It induces the second relaxation stage, where X_2_ species accumulated inside the membrane during the first stage are transformed at the electrode/membrane interface, as well as transferred from the membrane to the external solution. Their concentration distribution, C(x,t), is given by the nonstationary Fick equation, Equation (4), in combination with initial condition (24) and boundary conditions (26) and (3). Its solution may be found again with the use of the Laplace transform:(27)$$C(x,t)=2\;C^{*}\;\sum_{n=1}^{\infty }(\pi n){}^{-1}\;\sin\;(\pi n\;x/L)\;\exp\;(-n^{2}\;\pi^{2}\;D_{C}\;t/L^{2})$$
where C* is defined by Equation (24). This expression results in the *exact* solution, Equation (21), for the nonstationary cathodic current distribution, where the steady-state current, I_ss_, is defined by Equation (11):(28)$$I_{C}(t)=2\;I_{\mathrm{ss}}\;\sum_{n=1}^{\infty }\exp\;(-n^{2}\;\pi^{2}\;D_{C}\;t/L^{2})$$
which may be transformed again to another (also *exact*) expression by means of the Poisson summation formula:(29)$$I_{C}(t)=I_{\mathrm{ss}}\;\{-1+L\;(\pi t\;D_{C}){}^{-1/2}\;[1+2\;\sum_{k=1}^{\infty }\exp\;(-k^{2}\;L^{2}/D_{C}\;t)]\}$$

Integration of the nonstationary current in Equation (21) gives formulae for the total reduction charge, Q_C_: (30)$$\begin{array}{c} Q_{C}\;\equiv \;\int_{0}^{\infty }I_{C}(t)\;\mathrm{dt}=(1/3)\;I_{\mathrm{ss}}\;L^{2}/D_{C}=(1/3)\;(n/n_{C})\;\mathrm{FA}\;C^{*}\;L\\ =(1/3)\;(n/n_{B})\;\mathrm{FA}\;L\;K\;c^{0}\;D_{B}/D_{C} \end{array}$$
where n/n_C_ = 2 and n/n_B_ = 1 for reaction (1).

Expressions (29) and (21) provide *approximate* results for the current variation within the time ranges of its relatively small or large values, Equations (31), (32) or (34), respectively:I_C_(t) ≅ I_C(CC)_(t) = CC_C_/t^1/2^ − I_ss_ = 3 Q_C_ [(D_C_/πt L^2^)^1/2^ − D_C_/L^2^], i.e.,(31)
t^1/2^ I_C_(t) ≅ CC_C_ − I_ss_ t^1/2^, i.e., t^1/2^ I_C_(t)/I_ss_ ≅ L/(π D_C_)^1/2^ − t^1/2^(32)
for sufficiently small time values, where CC_C_ = lim {t^1/2^ I_C_(t)} (for t → 0), so that: CC_C_ = (n/n_C_) FA C* (D_C_/π)^1/2^ = CC_B_ (D_B_/D_C_)^1/2^ = I_ss_ L/(π D_C_)^1/2^;(33)
I_C_(t) ≅ 2 I_ss_ exp (−π^2^ D_C_ t/L^2^) = 6 Q_C_ D_C_ L^−2^ exp (−π^2^ D_C_ t/L^2^)(34)
for sufficiently large time values, where C* and I_ss_ are defined by Equations (24) and (11).

One should emphasize that Equations (31) and (32) are *more complicated than the Cottrell formula*, Equation (19), which takes place for the previous stage of the reaction of B species (X^−^ oxidation). Their difference originates from the different initial concentration distributions: the Cottrell evolution of the B (X^−^) concentration starts from its *uniform* distribution, Equation (6), while the distribution of C species develops from the *steady-state* concentration profile, C_ss_(x), in Equation (6), corresponding to the *direct current passage*, where the concentration of the reagent at this stage (X_2_ reduction) *varies* across the membrane. Nevertheless, the first term in Equations (31) and (32) is *identical* to the Cottrell expression for this process, where the bulk solution concentration of C species (which does not exist in this system after the stage of X^−^ reduction) is replaced by its value at x = 0, i.e., *inside the membrane at its boundary with the electrode surface*, C_ss_(0) = C*, Equations (31) and (32). The extra multiplier, n/n_C_, which is equal to 2 for X_2_ reduction, reflects the number of electrons transferred across the boundary per one reacting C (X_2_) species, Equation (1). Variation of the *initial* concentration distribution in space, C_ss_(x) in Equation (6), gives an extra term in Equations (31) and (32), which leads to *disappearance of the extended Cottrell plateau for the product*, I_C_(t) t^1/2^ in Equation (32), as a function of time, which is predicted by Equation (19) for the analogous product, I_B_(t) t^1/2^, at the first (X^−^ oxidation) stage.

These distinct features of two stages of the process are clearly visible in our experimental data, see below.

The long-time behavior, Equation (34), is given by an *exponential function of time*, which looks quite identical to approximate Equation (15) for the X^−^ oxidation, except for different diffusion coefficients, D_C_ in Equation (34) and D_B_ in Equation (15). However, it should be kept in mind that Equation (15) is valid for the *deviation of the current from its steady-state value*, I_ss_, while this behavior in Equation (34) is valid *for the current itself* since it *vanishes* for the long-time interval. This difference in predictions for the extended time range is of importance to consider over the course of the experimental data treatment, see below.

Thus, the use of experimental data for this X_2_ reduction stage allows one to extract the Q_C_ value as well as dependences I_C(CC)_(t) and I_C_(t), for the short or extended time range, respectively. Then, one can determine two independent parameters, e.g., FA C* L and L^2^/D_C_ or FA C* D_B_^1/2^. In combination with the information derived from the X^−^ reduction stage, one can directly determine the value of the diffusion coefficient of X_2_ (C) species, D_C_, as well as the ratio of the diffusion coefficients of X^−^ and X_2_ species, D_B_/D_C_.

## 4. Results and Discussion

### 4.1. Stationary Voltammograms

Figure 2 presents experimentally measured stationary voltammograms for the membrane-coated electrode in contact with a series of 2 M H_2_SO_4_ + x M HBr solutions for various x values from 0.125 to 1.0. For all plots, one can see a drastic increase of the steady-state current within the interval of potentials between 0.85 and 1.0 V due to bromide ion oxidation via reaction (1) for X = Br.

The plateau of the transport-limited current (slightly inclined region for 1 M HBr solution) is observed at more positive potentials. Figure 2b (black points) shows the dependence of the plateau current, I_plateau_, on the HBr concentration, where the uncertainty of its value for 1 M HBr due to the slope of the plateau is marked in Figure 2a. The I_plateau_ values are proportional to the HBr concentration within the range from 0.125 to 0.75 M, while the point for the 1 M solution deviates noticeably from this straight line.

Based on these results, non-stationary measurements of chronoamperograms in the diffusion-limited current regime for quantitative determination of the values of D and K parameters have been carried out at the potential step up to 1.0 V, see below.

### 4.2. Cyclic Voltammetry

Cyclic voltammograms for various potential scan rates, v, from 30 mV/s to 3 V/s (their values are indicated in the figures) are shown in Figure 3a for the 2 M H_2_SO_4_ + 0.25 M HBr composition of the external solution and in Appendix A for the whole set of HBr concentrations. In all cases, each CV plot possesses a single maximum for both forward and backward potential sweeps which reflects either the transformation of Br^−^ (oxidation) at the electrode surface into Br_2_ or a backward process (Br_2_ reduction) in accordance with the scheme given by Equation (1) for X = Br.

The theoretical description of the diffusional transport over the course of the CV experiment has been developed mostly for systems where the redox-active component is dissolved in a *semi-infinite* medium (which is usually considered as an electrolyte solution), while the transport takes place inside this medium to/from its boundary with the electrode. Results of such a consideration can be immediately applied to the problem of the diffusional transport inside a *spatially restricted* medium, e.g., inside a *layer of a finite thickness*, L, only if its value, L, is much larger than the thickness of the nonstationary diffusion layer, δ_D_. It should be kept in mind that the latter changes strongly in the course of the CV measurement: δ_D_(t), where δ_D_ is of the order of δ_D_(t_pf_) ~ (D RT/F v)^1/2^ within the initial time interval, which includes the range of the forward peak while it is extending during the further time interval as δ_D_(t) ~ [D (t − t_pf_)]^1/2^; in particular, it is about δ_D_(t_pb_) ~ (D t_pf-pb_)^1/2^ for the vicinity of the backward peak, where t_pf_ and t_pb_ are the time moments of passage of the forward or backward peaks, respectively, and t_pf-pb_ is the time for the passage between the forward and backward peaks of current: t_pf-pb_ = t_pb_ − t_pf_.

For the system under study, the diffusion coefficients of both transporting species, Br^−^ and Br_2_, inside the membrane are of the order of 10^−6^ cm^2^/s. Then, δ_D_(t_pf_) ~ 10 µm for the scan rate of a few tens of mV/s, while δ_D_(t_pb_) is greater by several times. It means that for the membrane thickness of 60 µm, this medium may be treated *as semi-infinite* for consideration of the shape of *the forward peak* (including the value of I_pf_) for *all studied scan rates* (including 30 mV/s), while the CV shape after the passage of the forward peak is described properly by the semi-infinite model *for scan rates of 100 mV/s and higher*, whereas it should be strongly modified for 30 mV/s. The correctness of this analysis is demonstrated in this section below.

One should emphasize that this reactional scheme corresponds to the *non-unity stoichiometry*, Equation (2), i.e., the stoichiometric numbers of its Red (Br^−^) and Ox (Br_2_) species are *different*: n_Red_ = 2, n_Ox_ = 1, and n = 2. According to the theory of such processes at the interface of an electrode and a *semi-infinite solution* under conditions of their *reversible* behavior [45], most of their *qualitative features* are *different* from those for conventional processes of the *unity stoichiometry*, Ox + n e^−^ = Red.

For example, the predicted difference between the peak potentials of the forward (oxidation) and backward (reduction) branches for the process given by Equation (2) (see Figure 1a and Table 1 of [45]) is around 43 mV for 25 °C, instead of 59 mV (n = 1) or 29.5 (n = 2) for the conventional redox processes. The experimental value, 44 mV, for the lowest HBr concentration, 0.125 M, and the optimal potential scan rate, 100 mV/s, in Figure 3a agrees perfectly with the theoretical one. An increase of the HBr concentration in the external solution leads (for the same scan rate) to greater values of the peak-to-peak separation, which may be related to the increase of the peak current and, correspondingly, of the shifts of both peak potentials (in opposite directions) due to ohmic effects.

For each HBr concentration, an increase of the potential scan rate results in *diminution of the peak-to-peak distance*. This tendency is opposite to that for redox reactions at the electrode/solution interface, where this peak-to-peak separation *increases* for higher scan rates because of ohmic effects. For the system under study, there is a *novel effect* which originates from a non-perfect morphology of the electrode/membrane interface where *lacunas of solution* are present (expectedly, due to the microroughness of the surfaces of the media in contact), which give an extra contribution to the current *within a very short time range* (this point is discussed in detail in the section below on chronoamperometry), thus shifting the backward peak to *earlier time values* and diminishing the peak-to-peak separation.

Another effect which is absent for reactions at conventional electrochemical interfaces manifests itself at the *lowest scan rate*, 30 mV/s, see the corresponding plot, e.g., in Figure 3c, where the current of each CV plot in Figure 3a has been divided by v^1/2^ so that the black line for 30 mV/s becomes well-visible. One may see an obvious difference of this plot from all those for higher scan rates within the potential range between the forward and backward peaks: instead of the current diminution in time according to the Cottrell law, the black line approaches *a constant value* corresponding to the *steady-state current*. It is in full agreement with the above estimation of the nonstationary diffusion layer thickness, which predicts the diffusion layer extension for such a scan rate up to the *whole membrane region*. As a result, the theory [45] derived for the semi-infinite solution region is *not applicable* for this scan rate within the potential range after passage of the forward peak. Due to this anomalous shape, the peak-to-peak separation for 30 mV/s and the 0.125 M HBr concentration (48 mV) becomes slightly larger than the prediction (43 mV).

According to the theory [45], variation of the potential scan rate, v, for a fixed reagent concentration should change the current as v^1/2^ for any value of the potential. The standard method to test this prediction is to plot the peak current for the forward scan, I_pa_, as a function of v^1/2^. One can see from Figure 3b that this proportionality between I_pa_ and v^1/2^ is observed for the sufficiently low scan rates, up to 300 mV/s, while there is a marked deviation for higher scan rates. One can also use a more advanced procedure which allows one to analyze this prediction within the whole range of potentials. It means that experimental CV data for various scan rates, v, should become *overlapping* if the current for each potential is divided by v^1/2^ (“normalized voltammograms”). Figure 3c presents such plots, I/v^1/2^ vs. E, for CV data in Figure 3a. One can see a practical coincidence of the plots in Figure 3c for various scan rates within the rising branch of the forward scan, which also takes place within the forward peak region for *lower scan rates* (up to 300 mV/s). The reason for the deviation of the black line for 30 mV/s between the forward and backward peaks has been discussed above, while the plots for the other scan rates are almost coincident within this potential range.

In conformity with Figure 3b, the plots for higher scan rates, 1 and 3 V/s, in Figure 3c show an *upward deviation* within the maximum region, while they are close to those for lower scan rates (except for that for 30 mV/s, see above) outside this potential range. One may attribute this effect to the observations over the course of *chronoamperometric* measurements (see the next section), where *upward* deviations from the Cottrell-type behavior have been found within *the shortest time range* (around 0.01 s and lower). Probably, this *excessive oxidation charge* is related to one of the above new effects: The existence of *solution-filled lacunas* at the electrode/membrane boundary manifests itself in the course of the cyclic voltammetry as *increasing intensity of the current near its peak*. Their contribution to the peak current becomes relatively significant at such *high scan rates* because of a small thickness of the nonstationary diffusion layer, in combination with the short duration of the peak passage.

Similar tendencies are observed for analogs of Figure 3b,c for the other HBr concentrations.

In conformity with theoretical predictions [45], the peak potential values for the forward scan, E_pa_, are practically independent of the scan rate (for the range from 0.03 to 1 V/s) for all HBr concentrations.

Predictions of the theory [45] on the dependence of CV plots on the bulk solution concentration of the reagent, c^0^ (shown in Figure 3d for the scan rate of 100 mV/s and in Appendix A for all scan rates), are essentially different, compared to those for conventional electron-transfer reactions, Ox + n e^−^ = Red. According to them, variation of the HBr concentration in the solution should lead not only to the proportional change of the peak-current intensities but also to a *shift of the whole plot* to the left for higher concentrations.

Figure 3e is prepared in conformity with the standard method to verify the proportionality prediction via plotting the peak current for the forward scan vs. the HBr concentration in the external solution, c^0^ (assuming the *equilibrium* HBr concentration inside the membrane, B*, is proportional to c^0^), where the straight line confirms the agreement with the prediction. More detailed information is provided in Figure 3e, which considers that the theory predicts this dependence on the concentration not only for the peak current but also for any potential. It is why the values of the CV current in Figure 3f are divided by the HBr concentration in the external solution, c^0^ (“normalized voltammograms” of another type). One can see from this figure that both principal predictions of the theory [45] have been confirmed: (1) increase of the concentration results in a *progressive shift to the left of the whole CV plot*, and (2) if these plots are shifted back to make the forward peak potentials superimposed, i.e., plotted vs. E—E_pa_, then they become very close to each other within the whole potential range, with a small variation of the amplitude along the Y-axis (which manifests itself as a slight deviation of points from the straight line in Figure 3e).

Thus, one can conclude on a good agreement of experimental CV data for the system under study with theoretical predictions [45] for redox reactions of the non-unity stoichiometry of the type given by Equation (2) as a whole. The best correspondence for the forward peak current, I_pa_, to Equation (5) is ensured by the lowest scan rates, 0.03 and 0.1 V/s. These values for I_pa_ have been used to calculate the CCV parameters, Equation (22), for various HBr concentrations. In the next section, they will be compared with the CC_B_ values for the same HBr concentrations found from chronoamperometric measurements. Then, these parameters will be used to determine the principal parameters of the system, D_B_ and K.

### 4.3. Chronoamperometry: X^−^ Oxidation Stage

During this stage, the electrode/membrane/solution system where bromide anions are uniformly distributed inside the membrane in equilibrium with the external solution is subjected to a large-amplitude potential step at t = 0 to a sufficiently positive potential value, which induces a rapid oxidation of species B (Br^−^) at the electrode surface so that their local concentration inside the membrane in the vicinity of the membrane/electrode boundary becomes close to zero. Temporal variation of the current (called “current transients” below), I_B_(t), within the time range: t > 0, after the potential step is shown for various HBr concentrations of the external solution, c^0^ (indicated inside the figure), in Figure 4a as well as in Figure 4b in the bilogarithmic coordinates, which allows one to analyze the behavior within the whole time range from 10^−4^ to 100 s.

According to Figure 4b, there are two characteristic time intervals where the behavior of current transients for the HBr concentrations between 0.125 and 0.75 M is totally different. Monotonously decreasing branches are visible within the *short-time range*, up to a few seconds. Within a *longer*-time range, from about 10 s, the current, I_B_(t), approaches a constant value, i.e., a steady-state current, I_ss_.

The current transient for the highest HBr concentration of the external solution, 1.0 M (Figure 4a), demonstrates *anomalous* features. In particular, the current passes *through a minimum* within the intermediate time range (about 6.7 s), followed by a *monotonous increase* of the current within a more extended range, without a well-expressed plateau behavior.

The values of I_ss_ (where the value of the current at the end of the measurement period is used as I_ss_ for the highest HBr concentration, c^0^ = 1.0 M) are plotted in Figure 2b as a function of the concentration, c^0^. One should note a very good agreement between the values of I_ss_ and I_plateau_ (found from the steady-state voltammetry) for the concentration interval between 0.125 and 0.75 M, as well as the proportionality of these values to the concentration (proximity to the straight line in Figure 2b). On the contrary, the values of I_ss_ and I_plateau_ for the highest HBr concentration are noticeably different. This point for I_ss_ in Figure 2b is located almost perfectly at the same straight line, i.e., it satisfies the proportionality property. However, the *sloped shape* of the current transient for this concentration (Figure 4a) within the time interval over 10 s provides strong evidence in favor of the conclusion that this coincidence is of an *accidental nature*, i.e., if the duration of this stage *were* different from the chosen value, 120 s, then this final value of the current would be quite different. One may also pay attention to the observation of the steady-state voltammetry for this concentration (Figure 2a) that *the plateau is also inclined*, i.e., one cannot find a *reliable* value for *the steady-state current*, while this information is necessary for the determination of the principal parameters, D_B_ and K. Moreover, the found values of the CC_B_ and CCV parameters for this concentration which characterize the short-time response (see below) are very far from each other, while the CC_B_ value is also needed for this calculation via Equation (23). Thus, we excluded the data for the highest concentration, 1.0 M, from the further analysis, in particular, from Figure 4c–f.

According to Equation (11), the steady-state current, I_ss_, depends on the product of three parameters, D_B_, K, and c^0^. One can see from Figure 2b that the proportionality of the steady-state current to the bulk solution concentration, c^0^, is confirmed for the HBr concentration range between 0.125 and 0.75 M. It implies that the product, D_B_ K, is practically independent of the HBr concentration.

To determine the values of another parameter of the process, CC_B_, defined by Equation (20), plots of Figure 4b have been represented in the form of the dependence, log (I_B_ t^1/2^) vs. log t (Figure 4c), where the Cottrell behavior of the current transient should give a constant value, CC_B_ ≅ I_B_ t^1/2^, Equation (19). Indeed, expected horizontal segments are evidently visible in Figure 4c within a broad time range (from 10^−3^ or 10^−2^ s to several seconds, depending on the HBr concentration), while the stages of the process both before and after this time range have distinctive features. In particular, the upward deviation from the horizontal line at relatively long-time values is evidently due to the gradual transition of the X^−^ concentration profile in the membrane and of the passing current from the Cottrell type, Equation (19), to the stationary (i.e., linear) concentration distribution throughout the membrane and to the steady-state current, Equation (11).

The deviation from the Cottrell behavior within the shorter time interval originated from processes that are not considered by the Cottrell model [46] deserves a more detailed analysis. Within the region of the shortest time interval (below 10^−3^ s), a time-independent current is observed (Figure 4b), which, moreover, does not depend on the HBr concentration. It is apparently unrelated to the bromide discharge. Within the subsequent time interval (of the order of 10^−3^–10^−2^ s), prior to the beginning of the Cottrell time range having the slope of ½ (from 10^−2^ s to several seconds), a sharp decrease of the current is observed in Figure 4b, where its value already significantly depends on the HBr concentration. Figure 4c shows that the product, I t^1/2^, passes through a *maximum* within this region.

To interpret the non-monotonous dependence of these experimental transients on time (for the coordinates of Figure 4c) for the membrane-coated electrode, similar measurements have been carried out for the uncoated electrode (without membrane) under identical conditions. The comparison of transients for these two systems (with and without membrane) in Figure 4c reveals that they *coincide* within the shortest time interval; then, from some time moment dependent on the HBr concentration, the bromide oxidation current at the membrane-coated electrode becomes much weaker than that without membrane for the same HBr concentration. One can also note that both the height of the maximum in Figure 4c and the time moment of its passage are shifted as a function of the HBr concentration.

These observations can be explained by the fact that microvolumes of the solution (*lacunae*) remain between the adjacent surfaces of the electrode and the membrane after assembly of the working electrode (Figure 1), due to their imperfect smoothness (depressions and protrusions). One should expect that the solution in the lacunae initially has the same concentration of the electroactive component, bromide, as the external solution, while the total surface area of the electrode in contact with the lacunae is much smaller than its total surface area, A. It allows us to explain both the dependence of the increasing branches of the curves in Figure 4c for the electrode/membrane system on the HBr concentration and significantly lower values of the current for it within this time range, compared to that for the uncoated electrode. For the electrode/solution system without membrane, the product, I·t^1/2^, for each HBr concentration in Figure 4c *immediately approaches a constant value*, CC_B_, Equation (19), for the solution after moving away from the universal straight line, while the corresponding graphs for the electrode/membrane/solution system drop sharply after passing their maxima. This decline due to the rapid depletion of bromide in the lacunae is followed by reaching a much smaller “Cottrell constant”, CC_B_, for the membrane in Figure 4c. Under this assumption, the total solution volume in these lacunae can be estimated from the amount of the excessive charge within the time interval when the plot in Figure 4c is above the horizontal CC_B_ line. Results of such an estimate, recalculated in terms of the effective thickness of the lacunae layer (as if it extends uniformly along the whole electrode surface), i.e., of the “gap” between the membrane and the electrode surfaces, do not exceed a few µm, i.e., they are comparable to the particle size of the abrasive used for polishing the metal. This coincidence testifies in favor of the proposed explanation of the non-monotonous behavior of current transients in Figure 4c within the short-time range.

Due to very small values of both the passing current and the duration of this time interval, the charge of this segment of the transient is very low, compared to that within the Cottrell region, especially in comparison with the total charge of the process. Therefore, the existence of this deviation does not significantly affect the treatment within this region and, consequently, results of further calculations of the principal parameters of the system, D_B_ and K, based on CC_B_ values, with the use of Equation (23), discussed below.

The CC_B_ parameter for each solution composition has been determined as the value of I_B_ t^1/2^ within the time range of *its constancy*, which extends in Figure 4c from around 0.01 s to several seconds. Values of the CCV parameter, i.e., an analog of the CC_B_ parameter, have also been found from CV data (peak current for the forward scan, I_pa_) in Figure 3 and Appendix A with the use of Equation (22) for each HBr concentration and each scan rate.

Figure 4d represents experimental data for CC_B_ and CCV for each external HBr solution as functions of the bulk solution HBr concentration, c^0^. One can conclude on the proximity of the CCV data for lowest scan rates, 0.03 and 0.1 V/s, between each other as well as to the corresponding CC_B_ value for the HBr concentrations from 0.125 to 0.75 M, while there is a marked upward shift of CCV values for larger scan rates, especially for 1 and 3 V/s (as a direct consequence of larger I_pa_/c^0^ values in Figure 3c) for all HBr concentrations.

One may note that the intensive oxidation of Br^−^ at the electrode under the CV regime only starts in the vicinity of the E_1/2_ potential, i.e., slightly before the current maximum, where the characteristic time of the Br^−^ consumption is around RT/Fv, i.e., below 0.01 s for v = 3 V/s and around 0.025 s for v = 1 V/s. It is the time range (below 0.01 or 0.02 s depending on the HBr concentration) where Figure 4c shows a *strong upward deviation from the Cottrell horizontal line*, attributed above as originated from a micrometer-thick lacunae layer. Thus, the larger CCV values for the two highest scan rates are related to the excessive current due to the oxidation of X^−^ species inside these lacunae. This is why our further analysis will be based on the CCV data for *the lowest scan rates*, 0.03 and 0.1 V/s, which are in a good agreement with independent CC_B_ data (Figure 4d) for all HBr concentrations up to 0.75 M.

According to the theoretical model, values of the CC_B_ and CCV parameters should be proportional to the product, K c^0^ D_B_^1/2^. Experimental data in Figure 4d as well as in Figure 4e (where the ratios of CC_B_ or CCV to c^0^ are practically independent of c^0^, especially for CC_B_ and for CCV at low scan rates) show a good proportionality between their values and the bulk solution HBr concentration, c^0^. It means that the product, K D_B_^1/2^, does not vary essentially within this range of c^0^. This conclusion matches well to that on another product, K D_B_, based on the above analysis of data for the steady-state current.

For each composition of the external solution, i.e., for each HBr concentration, c^0^, the shape of the theoretical chronoamperogram, I(t), depends on several parameters of the system, see Section 3.5 (X^−^ oxidation for scheme (1)). Their values are considered as known for some of them: F, R, T (293 °C), n = 2, n_B_ = 2, A (7.9 10^−3^ cm^2^), c^0^ (from 0.125 to 0.75 M), and L (58 µm). The values of two other parameters are to be found via a comparison of these theoretical predictions with experimental data for each HBr concentration given in Figure 4a: the diffusion coefficient, D_B_, of the reactive species, B (X^−^ for the system under consideration), inside the membrane, and the distribution coefficient for this species at its equilibrium across the membrane/solution boundary, K.

*Several independent procedures* (B1 to B3) have been developed in our study for determination of the D_B_ and K values for each c^0^ concentration. They are based on different experimental characteristics described by the relations derived above in the theoretical section so that the subsequent comparison of the obtained values allows one to enhance the precision of the results.

Procedure B1: It is based on expressions (20) for CC_B_ and (11) for I_ss_. Their values have been determined from experimental data, see Figure 4d and Figure 2b, and provided in column B1 of Table 1. This procedure combines data for very short and extended time ranges. Equation (23) provides values of both D_B_ and K, see column B1 in Table 2.

Procedure B2: It is based on calculations of the “nonstationary charge”, Q_B(ns)_, by integration of I_B(ns)_(t), i.e., of the deviation of the nonstationary current, I_B_(t), from the steady-state one, I_ss_, between 0 and the time range where they become very close to one another, Equations (16), (14) and (15). Effectively, the value of Q_B(ns)_ is found as the limit of the expression: the integral of I_B_(t) between 0 and t_ss_ minus the product, t_ss_ I_B_(t_ss_), for sufficiently large t_ss_ values. The values of Q_B(ns)_ for various HBr concentrations are provided in column B2 of Table 1. Then, the final expression for Q_B(ns)_ in Equation (16) immediately provides the values of K for various c^0^, as seen in column B2 of Table 2.

Procedure B3: It is based on the presentation of the data in Figure 4a in semilogarithmic coordinates, log [I_B(ns)_(t)/I_ss_] vs. t, for each HBr concentration. According to Equation (15), this plot for a sufficiently large time range should be close to a straight line passing through the origin. The value of the slope of this line, which should be equal to π^2^ D_B_/(2.30 L^2^), Equation (15), is indicated for each HBr concentration, c^0^, in column B3 of Table 1. In turn, it provides values of D_B_, see column B3 in Table 2.

Table 2 shows that various methods for finding the diffusion coefficient of bromide and its distribution coefficient between the membrane and the solution yield similar values for them for each HBr concentration. This self-consistency of the conclusions based on different treatment procedures, B1 to B3, has enabled us to conclude that the theoretical description used based on the nonstationary Fick equation provides a substantiated foundation for the bromide transport during the stage of its oxidation.

Moreover, the variation of values for each parameter as a function of the HBr concentration within the range from 0.125 to 0.75 M has turned out to be insignificant. Therefore, we have averaged all the results for each of these two parameters within the whole concentration range and obtained the following mean values: D_B_ = (2.98 ± 0.27)·10^−6^ cm^2^/s and K = 0.19 ± 0.005. 

### 4.4. Chronoamperometry: X_2_ Reduction Stage

Variation of the nonstationary current in time, I_C_ vs. t, after the second large-amplitude potential step (from 1.0 to 0.4 V) is shown in Figure 5a in the bilogarithmic coordinates (*decimal* logarithm is used again). Plots for the HBr concentrations in the external solution, c^0^, within the range between 0.125 and 0.75 M have approximately the same shape, being displaced along the *Y*-axis. This feature is in agreement with the theoretical prediction on the proportionality of the current to the concentration, c^0^, Equation (28) for I_C_ and Equation (11) for I_ss_. This behavior breaks down within the time range over 10 s since the intensity of the current becomes too weak for its reliable measurement.

The shape of current transients in Figure 5a depends on the same parameters given above for the stage of X^−^ oxidation as well as on *a single new parameter*, the diffusion coefficient of species C (X_2_) inside the membrane, D_C_. Similar to the preceding section, we have elaborated *several independent procedures* (C1 to C3) for treating experimental data in Figure 5a in order to determine the value of D_C_ for each HBr concentration, c^0^. Since they are based on the data inside different time ranges, the subsequent comparison of the values of D_C_ represents an important criterion of the applicability of the theoretical model and (if the model is found to be applicable) to increase the precision of the value of this transport parameter.

Procedure C1: It is based on the treatment of the transient data for the sufficiently short time range. As it is discussed in detail in the theoretical section, there is *no extended time range of the Cottrell behavior* because of the *spatial variation of the initial concentration profile*, C_ss_(x), Equation (24). This is why one has to use plots in the coordinates, t^1/2^ I_C_ vs t^1/2^, since Equation (32) predicts a *linear* dependence for them, with the slope equal to −I_ss_. Since the value of the steady-state current, I_ss_, has been found in the course of the previous stage (see Table 1), for each HBr concentration, c^0^, it is advantageous to plot the data for the short-time interval with the use of the coordinates, where the slope is equal to −1 and the intercept of the Y-axis determines the value of L/(π D_C_)^1/2^ (see Equation (32)). Experimental data plotted in Figure 5b are in perfect agreement with this prediction for c^0^ = 0.125 M, while the slope is slightly larger for the higher HBr concentrations. The values of the intercept, L/(π D_C_)^1/2^, found by the extrapolation of data in Figure 5b to the initial time moment, t = 0, are provided in column C1 of Table 3. The recalculated values of the diffusion coefficient, D_C_, in column C1 of Table 4 are very close to each other for all concentrations.

Procedure C2: The values of the total reduction charge, Q_C_, have been calculated via integration of the current transients in Figure 5a, Equation (30), for each HBr concentration. Their values are provided in column C2 of Table 3. According to Equation (30), the ratio of the steady-state current and of the X_2_ reduction charge, I_ss_/Q_C_ = 3 D_C_/L^2^, immediately gives the value of D_C_ for each HBr concentration, see column C2 of Table 4.

Procedure C3: It is based on the long-time behavior of current transients in Figure 5a. According to Equation (34), −log [I_C_/2 I_ss_] varies linearly in time, its slope being equal to π^2^ D_C_ t/(2.30 L^2^) for the decimal logarithm, while its extrapolation towards the initial time moment, t = 0, is equal to 0. Figure 5c demonstrates a confirmation of this prediction, especially for the lowest HBr concentration. Column C3 of Table 3 presents the value of the slope for each HBr concentration. One may conclude on a slight increase of its value for higher concentrations. Calculated values of D_C_ are provided in column C3 of Table 4.

Mean D_C_ values have been found by averaging those for various procedures of the data treatment, C1 to C3, in Table 4. They do not reveal a significant variation as a function of the HBr concentration in the external solution, c^0^, within the range under consideration. Therefore, the mean value of D_C_ for this concentration range has been determined by averaging the results for various concentrations: D_C_ = (1.10 ± 0.07) 10^−6^ cm^2^/s.

The values of the Br^−^ and Br_2_ transport parameters, D_B_ = D(Br^−^) = (2.98 ± 0.27) 10^−6^ cm^2^/s, D_C_ = D(Br_2_) = (1.10 ± 0.07) 10^−6^ cm^2^/s, and K = K(Br^−^) = 0.19 ± 0.005, within this range of HBr concentrations in the external solution (Table 2 and Table 4) show a good correlation with those obtained earlier for the transport of components of the bromine–bromide mixed solution across membranes of a similar type: D = 1.45 10^−6^ cm^2^/s and K = 0.29 [17,47]. It is hardly possible to expect a closer agreement of our results with the earlier ones since the quoted papers dealt with the *bromine* transport in the presence of *high bromide concentrations*, with the *tribromide* formation, so that it is not clear which of these components gave the main contribution to the diffusion inside the membrane. Besides, sulfuric acid was not added, unlike the case of our studies.

## 5. Conclusions

A novel method has been proposed for experimental studies of the crossover of a redox-active component of the external solution (B) as well as of the product (C) of its reaction at the electrode, which may be of *any stoichiometry*: n_B_ B + n e^−^ = n_C_ C, Equation (2), across an ion-exchange membrane. Unlike previous approaches towards this problem based on two-chamber cells and the corresponding experimental procedures, our method has allowed us to simplify essentially the measurement installation as well as to shorten the measurement duration and to diminish the size of the membrane sample. A special home-made installation has been elaborated where the working electrode is coated by an ion-exchange membrane so that a redox-active component of the external solution can only reach the electrode surface via its transport across the membrane. Variation of the electrode potential enables one to carry out various voltammetric and chronoamperometric studies, where the electroactive species, B, from the external solution as well as the product of its reaction at the electrode, C, have to cross the membrane.

The theoretical description of these processes is based on the nonstationary diffusion equations for these species, B and C, Equation (4), inside the membrane, while the process of their transfer across the membrane/solution boundary is assumed to be in equilibrium, despite the current passage.

Original analytical expressions for the nonstationary diffusion-controlled current over the course of the *two-step chronoamperometric regime* have been derived. Owing to them, *several different procedures* for the treatment of experimental data obtained by means of this technique have been proposed. As a result, the same transport parameter for species B or C may be estimated *in several independent ways*. Moreover, extra information on these parameters was also provided by *various voltammetric techniques*. The comparison of the results obtained via this set of experimental methods represents a key applicability criterion of the underlying theoretical model.

Experimental verification of these theoretical predictions has been performed for the transport of bromide anion, Br^−^, from an external aqueous solution (2 M H_2_SO_4_ + various amounts of HBr, from 0.125 to 1.0 M) across the Nafion-212 sulfonic cation-exchange membrane to the Pt electrode surface. Both the steady-state and nonstationary currents have been registered for voltammetric and chronoamperometric regimes.

Treatment of experimental data for these regimes based on our theoretical model, with the use of various treatment’s procedures, has allowed us to determine the diffusion coefficients inside the ion-exchange membrane for both B (X^−^) and C (X_2_) species as well as the distribution coefficient of species B (X^−^) at the membrane/solution interface for various HBr concentrations in the external solution, c^0^. Their values in the concentration range between 0.125 and 0.75 M were constant, within the experimental dispersion: D(Br^−^) = (2.98 ± 0.27) 10^−6^ cm^2^/s, D(Br_2_) = (1.10 ± 0.07) 10^−6^ cm^2^/s, and K(Br^−^) = 0.190 ± 0.005.

This possibility to determine transport characteristics of *two* redox species, a solute component, B, and the product of its redox transformation, C, *within a single experiment* represents a *unique feature* of this study.

The proposed method may also be useful for studies of transport characteristics of components of other redox couples. If the conditions of this process correspond to those in our study (the transmembrane transfer of electroactive species via their *molecular diffusion* as well as the *proportionality* between the surface concentrations of a species in the membrane and in the solution at their boundary), the proposed procedure should allow one to determine the transport characteristics. Otherwise, if the conditions assumed in our study are not satisfied, then the general methodology of our approach is still applicable, while one should use a more general theoretical description of the process to extract the crossover parameters.

## Figures and Tables

**Figure 1 membranes-12-01041-f001:**
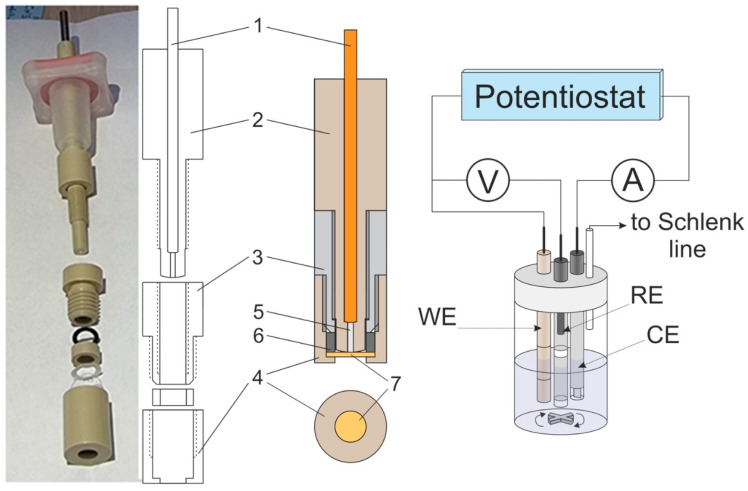
Photo of the working electrode for measuring transport parameters in ion-exchange membranes (**left**) and its scheme in disassembled and assembled states (in the middle), as well as a diagram of the three-electrode cell (**right**) used for measurements. Numbered elements: current supplier (1), body (2), membrane fixator (3), cover (4), platinum rod (5), sealing spacer (6), and membrane (7). Working electrode diameter: 1 mm.

**Figure 2 membranes-12-01041-f002:**
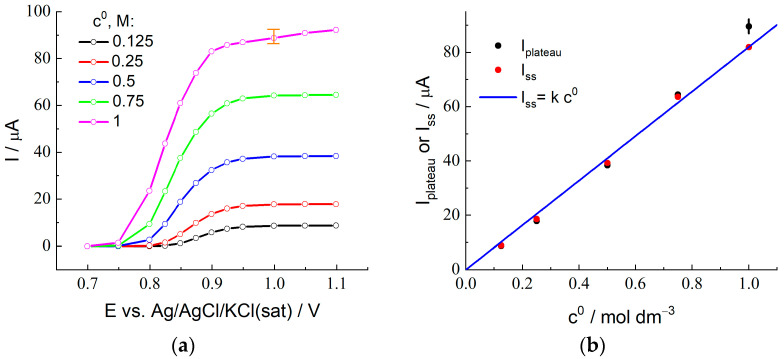
(**a**) Results of the stationary voltammetry for the platinum electrode (diameter: 1 mm) coated with Nafion 212 membrane in contact with 2 M sulfuric acid solution with the addition of HBr, c^0^ from 0.125 to 1.0 M (its values are indicated in the figure). (**b**) Dependence of the steady-state diffusion-limited current found as the plateau current of the stationary voltammograms, I_plateau_ (black), in (**a**), or as the stationary current on chronoamperograms, I_ss_ (red), in Section Chronoamperometry: X^−^ Oxidation Stage, on the HBr concentration in solution. The straight line (blue) given by the equation: I_ss_ = k c^0^, illustrates the proportionality of the stationary current, I_ss_, to the HBr concentration in solution.

**Figure 3 membranes-12-01041-f003:**
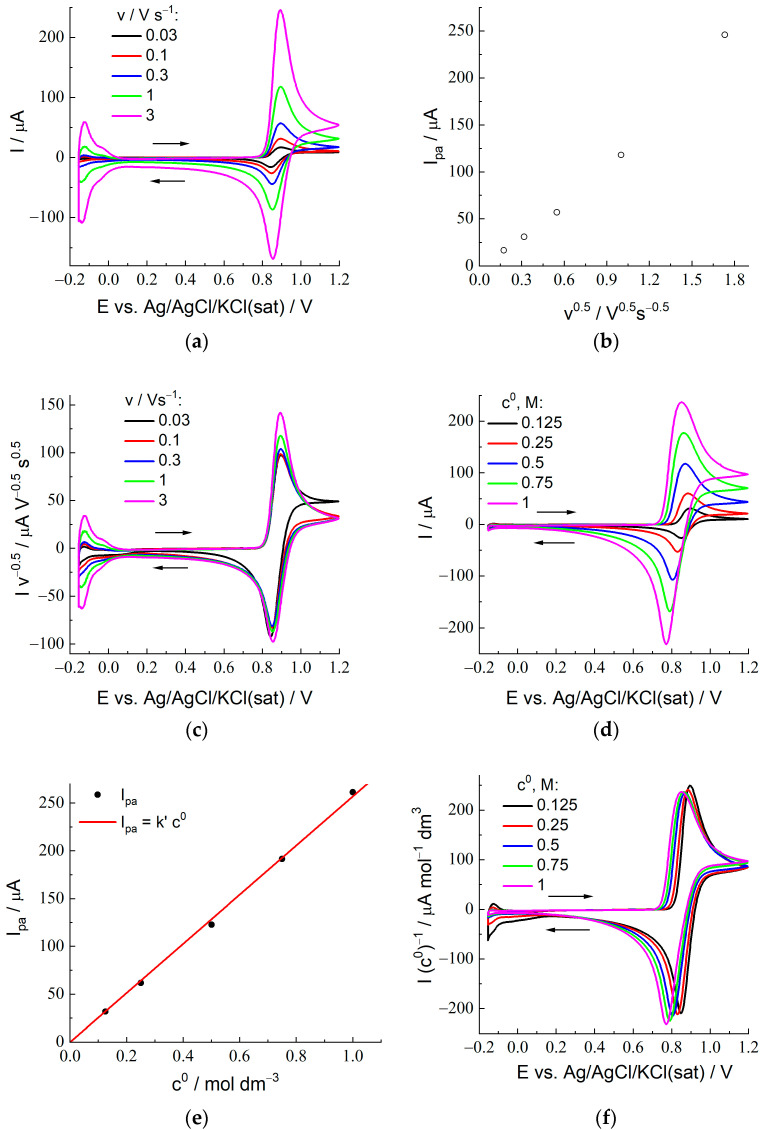
Cyclic voltammograms for the Pt electrode (diameter: 1 mm) coated with Nafion 212 membrane in contact with 2 M sulfuric acid with the addition of HBr, c^0^: (**a**) CV plots for a set of scan rates between 30 mV/s and 3 V/s (its values are indicated in the figure) for c^0^ = 0.125 M (data for all HBr concentrations are given in Appendix A). (**b**) Dependence of the forward (oxidation) peak current, I_pa_, on v^1/2^ for data of (**a**). (**c**) “Normalized CV plots”, i.e., dependences of I/v^1/2^ on E for various scan rates, data of (**a**). (**d**) CV plots for a set of the HBr concentrations, c^0^ (its values are indicated in the figure), for 100 mV/s (data for all scan rates are given in Appendix A). (**e**) Dependence of the anodic peak current of the voltammograms, I_pa_ (black), on the HBr concentration in the external solution, c^0^, in (**d**). The straight line (red), I_pa_ = k’ c^0^, illustrates the proportionality of the anodic peak current, I_pa_, to the HBr concentration in solution. (**f**) “Normalized CV plots”, i.e., dependences of I/c^0^ on E for various HBr concentrations, data of (**d**). See legend to Figure 2 for notations and values of parameters.

**Figure 4 membranes-12-01041-f004:**
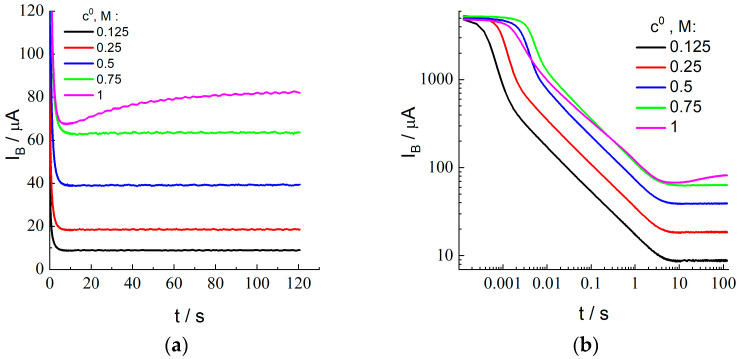
Chronoamperometry for the Pt electrode coated by Nafion-212 membrane in the course of the Br^−^ oxidation stage from aqueous 2 M H_2_SO_4_ solution with additions of various HBr concentrations, c^0^ (its values are indicated in the figure), after forward potential step from OCP to 1.0 V: (**a**,**b**) Dependence of the current, I_B_, on time in the linear (**a**) or in the bilogarithmic (**b**) coordinates (decimal logarithm is used everywhere in this paper). (**c**) Recalculation of data in (**a**,**b**) in the coordinates: log (I_B_ t^1/2^) vs. log t. Analogous results (dashed lines marked as “Pt”) for the Pt electrode of the same surface area under identical conditions, except for the absence of membrane coating. (**d**) Dependence of the Cottrell constant, CC_B_, Equation (20), and CCV on c^0^. CCV values are found from I_pa_ via Equation (22) for the set of scan rates (their values are indicated inside the figure) for the same HBr concentrations. (**e**) Data of (**d**) replotted for CC_B_/c^0^ or CCV/c^0^ vs. c^0^. (**f**) Dependence of log [I_B(ns)_(t)/(2 Iss)] on time, Equation (15), within the large-time range for various HBr concentrations.

**Figure 5 membranes-12-01041-f005:**
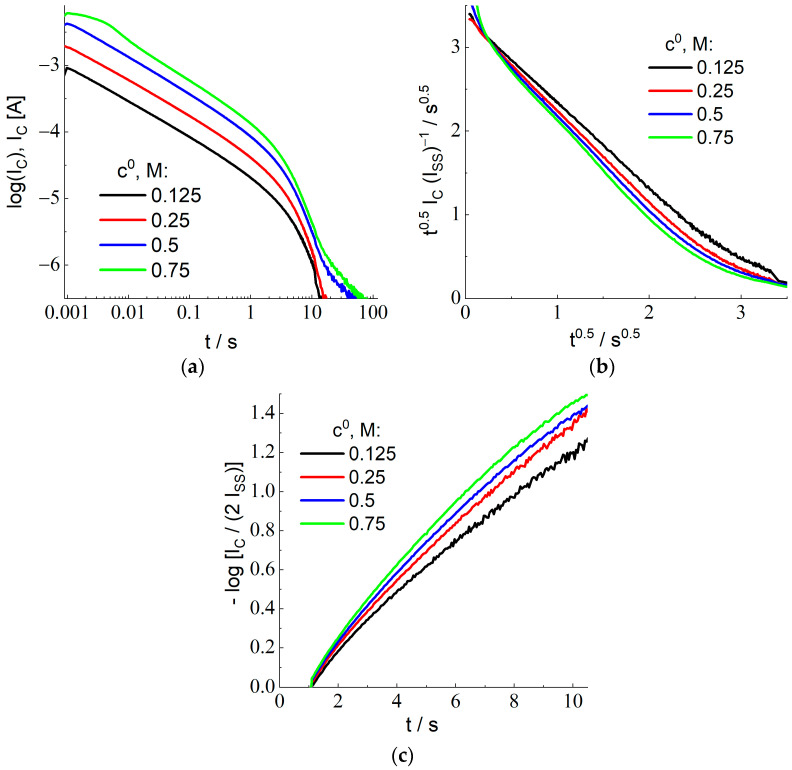
(**a**) Current transients after the backward potential step (from 1.0 to 0.4 V) for the Pt electrode (diameter: 1 mm) coated with the Nafion 212 membrane in contact with 2 M sulfuric acid with the addition of various HBr amounts, c^0^, from 0.125 to 0.75 M (its values are indicated in the figure) in the bilogarithmic (decimal logarithms) coordinates. (**b**) Experimental data of (**a**) within the short-time range recalculated with the use of the coordinates: t^1/2^ I_C_/I_ss_ vs t^1/2^, where the theory, Equation (32), predicts a linear dependence with the slope equal to −1 and with the intercept of the Y-axis equal to L/(π D_C_)^1/2^. (**c**) Experimental data of (**a**) within the extended-time range recalculated with the use of the coordinates: log [I_C_/I_ss_] vs. time, where the theory, Equation (34), predicts a linear dependence passing through the origin, its slope being equal to −π^2^ D_C_ t/L^2^.

**Table 1 membranes-12-01041-t001:** Results of various procedures, B1 to B3, for treatment of experimental current transients of the Br^−^ oxidation (Figure 4a).

c^0^, M	B1		B2	B3
	I_ss_	CC_B_	Q_B(ns)_	π^2^ D_B_/(2.30 L^2^)
	μA	μA s^0.5^	μC	s^−1^
0.125	8.8	16.9	35.3	0.326
0.25	18.4	34.6	69.9	0.337
0.5	39.2	72.2	142.3	0.354
0.75	63.0	113.6	229.3	0.301

**Table 2 membranes-12-01041-t002:** Values of the diffusion coefficient of Br^−^ inside the membrane, D_B_, and of its distribution coefficient, K, based on results for various procedures in Table 1, B1 to B3.

c^0^, M	B1		B2		B3
	D_B_·10^6^	K	K	D_B_·10^6^	D_B_·10^6^
	cm^2^/s	-	-	cm^2^/s	cm^2^/s
0.125	2.96	0.184	0.190	2.87	2.65
0.25	3.12	0.183	0.188	3.05	2.74
0.5	3.26	0.187	0.191	3.19	2.87
0.75	3.41	0.192	0.205	3.19	2.44
Mean	3.19	0.187	0.193	3.08	2.68
StdErr	0.19	0.004	0.008	0.15	0.18

**Table 3 membranes-12-01041-t003:** Results of various procedures, C1 to C3, for the treatment of experimental current transients for the Br_2_ reduction (Figure 5a).

c^0^, M	C1	C2	C3
	L/(πD_C_)^0.5^	Q^C^	−π^2^D_C_/(2.30 L^2^)
	s	μC	s^−1^
0.125	3.33	98.4	0.124
0.25	3.31	191	0.139
0.5	3.32	398	0.148
0.75	3.25	616	0.158

**Table 4 membranes-12-01041-t004:** Values of the diffusion coefficient of Br_2_ inside the membrane, D_C_, based on results for various procedures, C1 to C3, in Table 3.

c^0^, M	C1	C2	C3
	D_C_·10^6^	D_C_·10^6^	D_C_·10^6^
	cm^2^/s	cm^2^/s	cm^2^/s
0.125	1.00	1.03	1.01
0.25	1.01	1.12	1.13
0.5	1.01	1.14	1.20
0.75	1.05	1.19	1.28
Mean	1.02	1.12	1.16
StdErr	0.02	0.07	0.12

## Data Availability

Not applicable.

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
