# Peer review of "Electrochemical Measurement of Interfacial Distribution and Diffusion Coefficients of Electroactive Species for Ion-Exchange Membranes: Application to Br2/Br Redox Couple"

_membranes, 2022, doi:10.3390/membranes12111041_

Round 1

Reviewer 1 Report

The manuscript reported a new approach for the determination of fundamental parameters of the active species crossover process which is based on direct measurement of the current due to diffusion through the membrane of an electroactive component in solution and of its redox product with the use of a combination of voltammetric and chronoamperometric techniques. The stationary and cyclic voltammetry methods allow one to determine both the diffusion-limited steady-state current through the membrane and the Cottrell constant of the diffusion process. The possibility to determine transport characteristics of two redox species, the solute component and the product of its redox transformation within a single experiment¸ represents a unique feature of this study.

I consider the content of this manuscript will definitely meet the reading interests of the readers of the Membranes journal. However, there are certain English spelling and grammar issues, and also the discussion and explanation should be further improved.

Therefore, I suggest giving a minor revision and the authors need to clarify some issues or supply some more experimental data to enrich the content. This could be comprehensive and meaningful work after revision.

Detailed comments can be found in the PDF file.

Reviewer 2 Report

The cross-over phenomenon and the concentration of species at the interface especially in membrane surfaces are intricate questions for those working in the field of energy storage devices.    The authors have approached these two issues from both theoretical and experimental points of view.   They have constructed their own demonstration cell and the cell design is also provided.   They have studied the phenomenon by considering the conventional redox couple bromine/bromide redox couple.   However a few statements regarding the applicability of the principles developed for other redox couples would have been useful

In view of the importance of this study of the interface phenomenon and the ways of determining the concentrations at the interface are worth reporting.
